# Cycle-consistent Masked AutoEncoder for Unsupervised Domain Generalization

**Haiyang Yang**[1,5][*], **Xiaotong Li**[2][*], **Shixiang Tang**[3,5][†], **Feng Zhu**[5], **Yizhou Wang**[4,5]
**Meilin Chen**[4,5], **Lei Bai**[7], **Rui Zhao**[5,6], **Wanli Ouyang**[7]
[1]Nanjing University [2]Peking University [3]The University of Sydney [4]Zhejiang University
[5]Sensetime Research [6]Qing Yuan Research Institute [7]Shanghai AI Laboratory
hyyang@smail.nju.edu.cn, lixiaotong@stu.pku.edu.cn
baisanshi@gmail.com, {yizhouwang, merlinis}@zju.edu.cn
stan3906@uni.sydney.edu.au, {zhufeng, zhaorui}@sensetime.com
wanli.ouyang@sydney.edu.au

## Abstract

Self-supervised learning methods undergo undesirable performance drops when there exists a significant domain gap between training and testing scenarios. Therefore, unsupervised domain generalization (UDG) is proposed to tackle the problem, which requires the model to be trained on several different domains without supervision and generalize well on unseen test domains. Existing methods either rely on a cross-domain and semantically consistent image pair in contrastive methods or the reconstruction pair in generative methods, while the precious image pairs are not available without semantic labels. In this paper, we propose a cycle cross-domain reconstruction task for unsupervised domain generalization in the absence of paired images. The cycle cross-domain reconstruction task converts a masked image from one domain to another domain and then reconstructs the original image from the converted images. To preserve the divergent domain knowledge of decoders in the cycle reconstruction task, we propose a novel domain-contrastive loss to regularize the domain information in reconstructed images encoded with the desirable domain style. Qualitative results on extensive datasets illustrate our method improves the state-of-the-art unsupervised domain generalization methods by average **+5.59%**, **+4.52%**, **+4.22%**, **+7.02%** on 1%, 5%, 10%, 100% PACS, and **+5.08%**, **+6.49%**, **+1.79%**, **+0.53%** on 1%, 5%, 10%, 100% DomainNet, respectively.

## 1 Introduction

Recent progresses have shown the great capability of unsupervised learning in learning good representations without manual annotations (Doersch et al., 2015; Noroozi & Favaro, 2016; Gidaris et al., 2018; Chen et al., 2020b; He et al., 2020; Chen et al., 2021; Zbontar et al., 2021; Caron et al., 2021; Tian et al., 2020; Henaff, 2020; Oord et al., 2018; Wu et al., 2018; Misra & Maaten, 2020; Caron et al., 2020; Li et al., 2022; 2023). However, they mostly rely on the assumption that the testing and training domain should follow an independent and identical distribution. In many real-world situations, this assumption is hardly held due to the existence of domain gaps between the training set and testing set in the real world. As a result, significant performance drops can be observed when deep learning models encounter out-of-distribution deployment scenarios (Zhuang et al., 2019; Sariyildiz et al., 2021; Wang et al., 2021; Bengio et al., 2019; Engstrom et al., 2019; Hendrycks & Dietterich, 2018; Recht et al., 2019; Su et al., 2019). A novel setting, unsupervised domain generalization (UDG) (Zhang et al., 2022; Harary et al., 2021; Yang et al., 2022), is therefore introduced to solve the problem, in which the model is trained on multiple unlabeled source domains and expected to generalize well on unseen target domains.

---

[*]Equal Contribution
[†]Corresponding Author

Existing unsupervised domain generalization methods rely on constructing cross-domain but semantically consistent image pairs to design the pretext tasks, *i.e.,* contrastive-based (Zhang et al., 2022; Harary et al., 2021) and generative-based methods (Yang et al., 2022). The contrastive-based methods aim to push the cross-domain positive pairs (samples of the same classes but from different domains) together and pull the negative pairs (the samples of different classes) apart (Fig. 1(a)). In contrast, the generative-based method proposes a new cross-domain masked image reconstruction task to recover the original image based on its style-transferred counterpart, which aims to disentangle the domain information and obtain a domain-invariant content encoder (see Fig. 1(b)). Although they achieve great success in unsupervised domain generalization, how to fully exploit the multiple domain information and establish better input-target reconstructed pairs is still a fundamental challenge. In fact, the reconstructed pairs are expected to cover more realistic and diverse cross-domain sample pairs, but without image annotations, those pairs can not be accurately obtained.

To tackle the challenge above, we propose a generative-based model named **Cycle**-consistent **M**asked **A**uto**E**ncoder (**CycleMAE**). Our method designs a novel cycle cross-domain reconstruction task for unsupervised domain generalization in the absence of paired images, which reconstructs multiple images from different domains in a self-circulating manner with a masked autoencoder. Specifically, the cycle cross-domain reconstruction task is to reconstruct an image under randomly masking from one domain to another domain, and then bring this generated counterpart back to its original domain, forming the self-circulating approach (as illustrated in Fig. 1(c)). In this way, we can establish the cross-domain counterparts of multiple domains by the forward reconstruction, and construct high-quality input-target pairs for the backward reconstruction. Since the high-quality input-target pairs are from the model outputs, they can be more realistic than cross-domain pairs designed by manual rules. With these more realistic pairs, the models can be taught to better disentangle the domain-invariant features.

We further observe that directly applying the cycle reconstruction tasks may underestimate the model's ability to disentangle the style information in the domain-specific decoders. Without any supervision of the reconstructed images in the forward reconstruction, the model tends to take "shortcuts" that domain-specific decoders generate images in a similar domain to reduce the learning difficulty of encoders to extract content information in the backward reconstruction. To this end, we additionally introduce a domain contrastive loss to keep different decoders capturing divergent information. Specifically, we accomplish this regularization by pulling the samples with the same domain labels close and pushing the samples with different domain labels apart, which can force the decoder to capture less redundant information from each other and thus preferably disentangle the domain information.

To demonstrate the effectiveness of CycleMAE, massive experiments are conducted on the commonly used multi-domain UDG benchmarks, including PACS (Li et al., 2017) and DomainNet (Peng et al., 2019). The experiment results demonstrate that CyCleMAE achieves new state-of-the-art and obtains significant performance gains on all correlated unsupervised domain generalization settings (Zhang et al., 2022). Specifically, CycleMAE improves the states-of-the-art unsupervised domain generalization methods by Average **+5.59%**, **+4.52%**, **+4.22%**, **+7.02%** on 1%, 5%, 10%, 100% data fraction setting of PACS, and **+5.08%**, **+6.49%**, **+1.79%**, **+0.53%** on 1%, 5%, 10%, 100% data fraction setting of DomainNet.

Our contributions are two-fold: (1) We propose a self-circulating cross-domain reconstruction task to learn domain-invariant features. (2) We propose a domain contrastive loss to preserve the domain discriminativeness of the transformed image in the cycle cross-domain reconstruction task, which regularizes the encoder to learn domain-invariant features. Extensive experiments validate the effectiveness of our proposed method by improving the state-of-the-art generative-based methods by a large margin. Related works will be elaborated on in the Appendix 5.1.

## 2 Cycle-consistent Masked Autoencoder

We now introduce our cycle-consistent Masked Autoencoder (CycleMAE) to learn domain-invariant features for unsupervised domain generalization based on a simple generative baseline method DiMAE. Given images in a set of $N$ different domain $\mathbb{X} = \{\mathbb{X}_1, \mathbb{X}_2, ..., \mathbb{X}_N\}$, our proposed CycleMAE consists of a transformer-based content encoder $\mathcal{E}$ and multiple domain-specific transformer-based decoders $\mathcal{D} = \{\mathcal{D}_1, \mathcal{D}_2, ..., \mathcal{D}_N\}$, where $N$ is the number of domains in the training dataset and $\mathcal{D}_i$

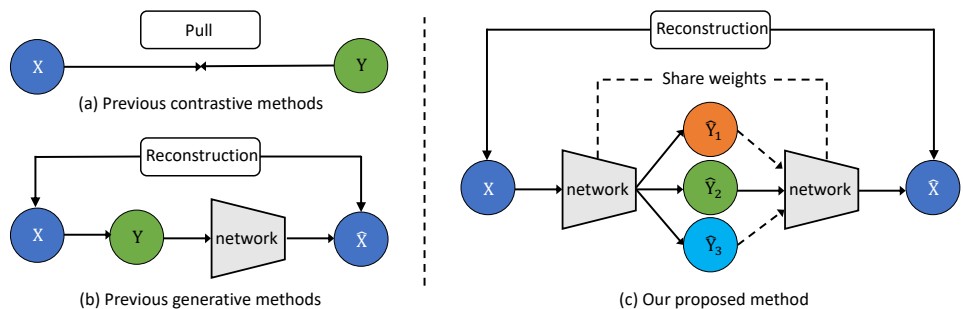

Figure 1: Comparison of previous and the proposed methods for UDG task. (a): Contrastive methods aim at pulling cross-domain but semantic similar images together. (b): Previous generative-based methods aim at reconstructing the original image based on the cross-domain images (generated by handcraft style-mix manners). (c): Our proposed CycleMAE leverages a self-circulating cross-domain reconstruction task for unsupervised domain generalization in the absence of paired data. (X and Y are a cross-domain but semantically the same image pair.)

is the domain-specific decoder of the image domain $\mathbb{X}_i$. The key innovation of our proposed method lies in easing the difficulty of constructing the cross-domain but semantically same image pairs by proposing a novel cycle cross-domain reconstruction task and a domain contrastive loss. Given an image $\mathbf{x}$ from the domain $\mathbb{X}_i$, the image reconstruction cycle should be able to bring $\mathbf{x}$ back to the original image, *i.e.,*

$$\mathbf{x} \to \mathbf{y} = \mathcal{D}(\mathcal{E}(\mathbf{x})) \to \hat{\mathbf{x}} = \mathcal{D}_i(\mathcal{E}(\mathbf{y})) \approx \mathbf{x}, \tag{1}$$

where $\mathbf{y} = \{\mathbf{y}_1, \mathbf{y}_2, ..., \mathbf{y}_N\}$ is the images generated by the forward transformation, $\mathbf{y}_i = \mathcal{D}_i(\mathcal{E}(\mathbf{x}))$ is the generated image encoded with the style of $\mathbb{X}_i$, and $\hat{\mathbf{x}}$ is the reconstructed image in $\mathcal{X}_i$ after the cycle cross-domain reconstruction task. The domain contrastive loss is used to preserve the domain difference among $\mathbf{y}$ to regularize the domain-specific decoder to learn domain style information.

## 2.1 OVERVIEW OF CYCLEMAE

As shown in Fig. 2, the proposed CycleMAE is based on DiMAE (Yang et al., 2022), but differently undergoes two consecutive and reversible processes: the forward transformation process (illustrated by blue arrows) and the backward transformation process (illustrated by green arrows). For the forward transformation (Step 1), we mostly follow the process in DiMAE, which transforms a style-mixed image to images from different domains, *i.e.,* $\mathbf{x} \to \mathbf{y}$. For the backward transformation process (Step 2), we transform the generated images $\mathbf{y}$ to $\hat{\mathbf{x}}$ to reconstruct $\mathbf{x}$. The cycle consistency loss, the domain contrastive loss along with the original cross-domain reconstruction loss in DiMAE are imposed in Step 3.

*Step1: Transform an image* $\mathbf{x}$ *to the forward-transformed images* $\mathbf{y}_1, \mathbf{y}_2, ..., \mathbf{y}_N$ *(blue arrow).* Given an image $\mathbf{x}$ in $\mathbb{X}_i$, we implement style-mix in (Yang et al., 2022) to generate its style-mixed image $\mathbf{v}$. Then we randomly divide $\mathbf{v}$ into visual patches $\mathbf{v}_v$ and masked patches $\mathbf{v}_m$. We feed the visual patches $\mathbf{v}_v$ into the encoder-decoder structure to generate the forward-transformed images $\mathbf{y} = \{\mathbf{y}_1, \mathbf{y}_2, ..., \mathbf{y}_N\}$.

*Step 2: Transform the generated images* $\mathbf{y}$ *to the image* $\hat{\mathbf{x}}$ *(green arrow).* Given a set of forward-transformed images $\mathbf{y}$, we randomly divide each of them into visual patches $\mathbf{y}_v$ and mask patches $\mathbf{y}_m$. Then we use the visual patches $\mathbf{y}_v$ to reconstruct the image $\hat{\mathbf{x}}$, where $\hat{\mathbf{x}}$ and $\mathbf{x}$ belong to the same domain.

*Step 3: Network optimization with the cross-domain reconstruction loss, the proposed domain constructive loss, and the cycle reconstruction loss.* The parameters in the encoder $\mathcal{E}$ and domain-specific decoders $\mathcal{D} = \{\mathcal{D}_1, \mathcal{D}_2, ..., \mathcal{D}_N\}$ are optimized by the cycle-reconstruction loss (Eq. 4), the domain contrastive loss (Eq. 5) and the cross-domain reconstruction loss (Eq. 6) used in (Yang et al., 2022).

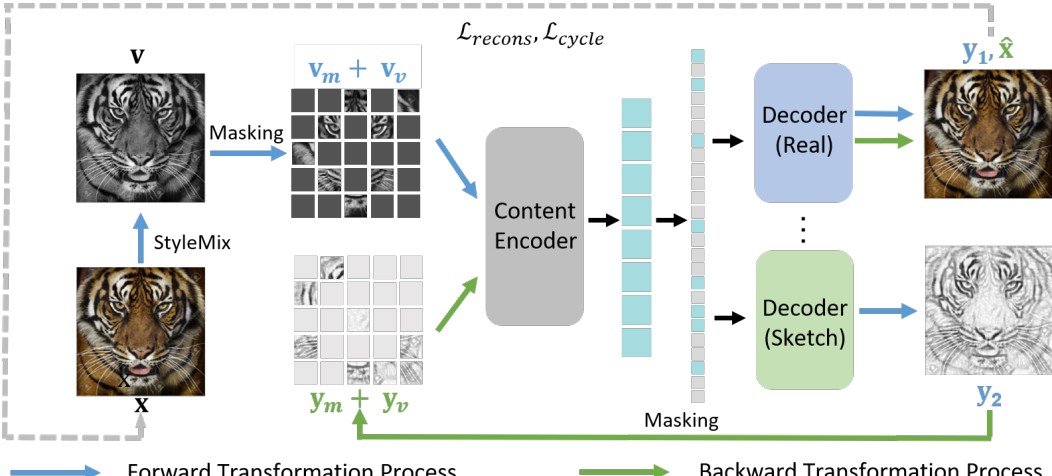

Figure 2: The illustration of CycleMAE with the cross-domain Reconstruction task. The training process includes the forward and backward transformation process. For an image $\mathbf{x}$, the forward transformation process transforms it to reconstruct the images in multiple domains. The backward transformation process then brings them to the original domain. We use the cross-domain reconstruction loss, the cycle cross-domain reconstruction loss and the domain contrastive loss to supervise the model optimization.

## 2.2 CYCLE CROSS-DOMAIN RECONSTRUCTION TASK

Previous contrastive-based methods and generative-based methods rely on high-quality cross-domain but semantically different image pairs to construct the pretext task. However, we argue that typical algorithms, *e.g.,* using different image augmentations on the same image or selecting nearest neighbors, can not preciously define such paired images because of the large domain gap. Therefore, we propose the cycle cross-domain reconstruction task to generate the cross-domain and semantic similar image pairs in a self-circulating way, *i.e.,* transforming an image to other domains and then bringing the transformed images to their original domain. Specifically, the overall cycle cross-domain reconstruction task consists of the forward transformation process and backward transformation process. The cross-domain cycle reconstruction loss is utilized to minimize the distance between the generated image after the cycle process and the original image.

**Forward transformation process.** Forward transformation process transforms an image into images in multiple domains with the designed encoder and domain-specific decoder. Specifically, given an image $\mathbf{x}$ in the domain $\hat{\mathbb{X}}$, we leverage the style-mix in (Yang et al., 2022) to the style-mixed image $\mathbf{v}$, and then randomly divide them into the visual patches and masked patches $\mathbf{v}_v$ and $\mathbf{v}_m$. The visual patches will be fed into the encoder $\mathcal{E}$ to extract the content feature $\mathbf{z}$, *i.e.,*

$$\mathbf{z} = \mathcal{E}(\mathbf{v}_v). \tag{2}$$

With the domain-specific decoders $\{\mathcal{D}_1, \mathcal{D}_2, ..., \mathcal{D}_N\}$, and the masked patches query $\mathbf{q}$, and the content feature $\mathbf{z}$, we reconstruct the images $\mathbf{y} = \{\mathbf{y}_1, \mathbf{y}_2, ..., \mathbf{y}_N\}$ in multiple domains $\mathbb{X} = \{\mathbb{X}_1, \mathbb{X}_2, ..., \mathbb{X}_N\}$. Given a masked query $\mathbf{q}$, the reconstructed image $\mathbf{y}_i$ is defined as $\mathbf{y}_i = \mathcal{D}_i(\mathbf{z}, \mathbf{q})$.

**Backward transformation process.** Backward transformation process transforms the generated images $\{\mathbf{y}_1, \mathbf{y}_2, ..., \mathbf{y}_N\}$ from multiple domains to the image $\hat{\mathbf{x}}$ in the domain of the original image to reconstruct the original image $\mathbf{x}$. Specifically, we divide every image $\mathbf{y}_i$ into the visible patches $\mathbf{y}_i^v$ and masked patches $\mathbf{y}_i^m$, where $i = 1, 2, ..., N$. We follow the implementation in forward transformation by replacing the $\mathbf{v}_v$ with $\mathbf{y}_i^v$, *i.e.,* $\mathbf{t}_i = \mathcal{E}(\mathbf{y}_i^v)$. With the decoder $\hat{\mathcal{D}}$ where $\hat{\mathcal{D}}$ generates images in the same domain $\hat{\mathbb{X}}$ as $\mathbf{x}$, a set of masked patches queries $\{\mathbf{q}^1, \mathbf{q}^2, ..., \mathbf{q}^N\}$ for $\{\mathbf{y}_1, \mathbf{y}_2, ..., \mathbf{y}_N\}$, and the content feature $\{\mathbf{t}_1, \mathbf{t}_2, ..., \mathbf{t}_N\}$, we reconstruct the image $\hat{\mathbf{x}}$ in $\mathbb{X}$. Mathematically, given masked queries $\{\mathbf{r}_1, \mathbf{r}_2, ..., \mathbf{r}_N\}$, the reconstructed image $\hat{\mathbf{x}}_i$ from $\mathbf{t}_i$ can be formulated as

$$\hat{\mathbf{x}}_i = \hat{\mathcal{D}}(\mathbf{t}_i, \mathbf{r}_i). \tag{3}$$

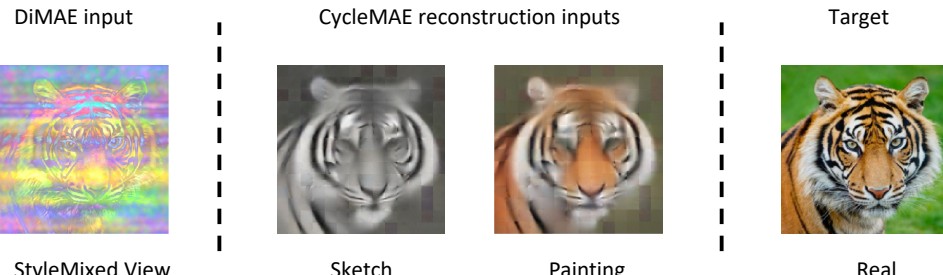

Figure 3: DiMAE leverages the heuristic methods in (Yang et al., 2022) to construct the reconstruction input with lots of artifacts. The CycleMAE utilizes the cross-domain reconstructed images as the reconstruction input, which are more natural and realistic.

**Cycle reconstruction loss.** The cycle reconstruction loss minimizes the distance between the masked patches and the corresponding masked patches of the original image $\mathbf{x}$. Specifically,

$$\mathcal{L}_{cycle} = \sum_{i=1}^{N} (\hat{\mathbf{x}}_i - \mathbf{x})^2, \tag{4}$$

where $\hat{\mathbf{x}}_i$ are the outputs of the backward transformation process and $\mathbf{x}$ is the original image.

**Discussion.** The high quality of the input-target reconstruction pair should be both natural and diverse. As shown in Fig. 3, the previous method, *i.e.,* DiMAE, constructs the input-target pair by the heuristic style-mix method, which causes artificial defects in the reconstruction input. In contrast, our CycleMAE leverages images generated by domain-specific decoders as the inputs for reconstructing the original image. Our reconstruction inputs do not rely on the heuristic designs and utilize the information in the deep model, which is more natural. Furthermore, owing to the multiple domain-specific designs of the generative-based UDG method, we can generate multiple and diverse inputs, instead of one image in DiMAE, for reconstructing the original image.

## 2.3 DOMAIN CONTRASTIVE LEARNING

Although the cycle cross-domain reconstruction task can generate precious input-target reconstruction pairs by the self-circulating process, we observe that directly applying the cycle reconstruction tasks decreases the model's ability to disentangle the style information in domain-specific decoders, which will underestimate the ability of the encoder to learn domain-invariant features. Without any supervision of the transformed images in the forward transformation process, the model tends to take "shortcuts" that domain-specific decoders generate images in a similar domain to reduce the difficulty of the encoder to extract content information in the backward transformation process.

Therefore, we propose a domain contrastive loss to regularize different decoders to learn divergent domain style information, which could give the encode diverse inputs for the encoder to reconstruct the original image in the backward transformation process. Specifically, domain contrastive learning loss aims at pulling the samples in the same domain close and pushing the samples in different domains apart. As mentioned in (Cao et al., 2022), the semantic information of features increases as the layers go deeper. To effectively regularize the decoder to learn divergent domain information and minimizes the influence on the semantic information, we utilize the features $\mathbf{d}$ from the first decoder transformer layer for domain contrastive learning, where $\mathbf{d}_i = \mathcal{D}_i^1(\mathbf{z}, \mathbf{q})$. Here, $\mathcal{D}_i^1$ denotes the first transformer layer of $\mathcal{D}_i$. Given an intermediate feature $\mathbf{d}_i$ of the decoder and the content feature $\mathbf{z}$ defined in Eq. 2, the domain contrastive loss is defined as

$$\mathcal{L}_{domain} = -\sum_{i=1}^{N} \log \frac{\exp(\mathbf{d}_i \cdot \mathbf{d}_i^+/\tau)}{\exp(\mathbf{d}_i \cdot \mathbf{d}_i^+/\tau) + \exp(\mathbf{d}_i \cdot \mathbf{d}_i^-/\tau)}, \tag{5}$$

where $\mathbf{d}_i^+$ denotes the features of samples in the same domain with $\mathbf{d}_i$ and $\mathbf{d}_i^-$ denotes the features of samples in the different domains of $\mathbf{d}_i$ among the training batch. With the domain contrastive loss, the decoder can generate images with divergent domain styles, giving diverse input images for the encoder to reconstruct the original images in the backward transformation process.

## 2.4 OBJECTIVE FUNCTION

Our cycle cross-domain reconstruction task can be compatible to the original cross-domain reconstruction task in (Yang et al., 2022). Therefore, to take the most advantage of reconstruction tasks, we also preserve the original cross-domain reconstruction loss proposed in (Yang et al., 2022) in our total objective function. Specifically, given an image from the domain $\mathbb{X}_i$, the cross-domain reconstruction is formulated as

$$\mathcal{L}_{recons} = (\mathbf{y}_i - \mathbf{x})^2, \tag{6}$$

Therefore, our total objective function can be formulated as

$$\mathcal{L} = \mathcal{L}_{recons} + \alpha \mathcal{L}_{domain} + \beta \mathcal{L}_{cycle} \tag{7}$$

where $\alpha$ and $\beta$ are hyperparameters that can be empirically set to 2 and 2. The sensitivity analysis of hyperparameters is presented in the Appendix 5.2.1.

## 3 EXPERIMENTS

### 3.1 EXPERIMENTAL SETUP

**Dataset.** Two benchmark datasets are adopted to carry through these two settings. PACS, proposed by (Li et al., 2017), is a widely used benchmark for domain generalization. It consists of four domains, including Photo (1,670 images), Art Painting (2,048 images), Cartoon (2,344 images), and Sketch (3,929 images) and each domain contains seven categories. (Peng et al., 2019) proposes a large and diverse cross-domain benchmark DomainNet, which contains 586,575 examples with 345 object classes, including six domains: Real, Painting, Sketch, Clipart, Infograph, and Quickdraw.

Following the all correlated setting of DARLING (Zhang et al., 2022), we select Painting, Real, and Sketch as source domains and Clipart, Infograph, and Quickdraw as target domains for DomainNet. In this setting, we select 20 classes out of 345 categories for both training and testing, exactly following the setting in (Zhang et al., 2022). For PACS, we follow the common setting in domain generalization (Li et al., 2018; Rahman et al., 2020; Albuquerque et al., 2019) where three domains are selected for self-supervised training, and the remaining domain is used for evaluation. Except above, the remaining experiments will be shown in Appendix 5.2.2.

**Evaluation protocol.** We follow the all correlated setting of DARLING (Zhang et al., 2022) and divide the testing process into three steps. First, we train our model in the unlabeled source domains. Second, we use a different number of labeled training examples of the validation subset in the source domains to finetune the classifier or the whole backbone. In detail, when the fraction of the labeled finetuning data is lower than 10% of the whole validation subset in the source domains, we only finetune the linear classifier for all the methods. When the fraction of labeled finetuning data is larger than 10% of the whole validation subset in the source domains, we finetune the whole network, including the backbone and the classifier. Last, we can evaluate the model on the target domains.

**Implementation details.** We use ViT-small as the backbone network unless otherwise specified. The learning rate for pre-training is $1.5 \times 10^{-5}$ and then decays with a cosine decay schedule. The weight decay is set to 0.05 and the batch size is set to $256 \times N_d$, where $N_d$ is the number of domains in the training set. All methods are pre-trained for 1000 epochs, which is consistent with the implementations in (Zhang et al., 2022) for fair comparisons. The feature dimension is set to 1024. For finetuning, we follow the exact training schedule as that in (Zhang et al., 2022). We use a MAE (He et al., 2021) unsupervised pre-training model in ImageNet for 1600 epochs to ensure labels are not available during the whole pretraining process.

### 3.2 EXPERIMENTAL RESULTS

We present the results in Tab. 8 (DomainNet) and Tab. 9 (PACS), which shows that our CycleMAE achieves better performances on most tasks compared to previous methods. Specifically, CycleMAE improves the performance by **+3.81%** and **+5.08%** for DomainNet on overall and average accuracy for 1% fraction setting. For 5% fraction setting, CycleMAE improves the previous methods by **+5.53%** and **+6.49%** for DomainNet on overall and average accuracy. For 10% and 100% fraction settings, which adopts finetuning for the whole network, CycleMAE improves the state-of-art by

| Methods | Label Fraction 1% (Linear evaluation) | | | | | Label Fraction 5% (Linear evaluation) | | | | |
|---|---|---|---|---|---|---|---|---|---|---|
| | Clipart | Infograph | Quickdraw | Overall | Avg. | Clipart | Infograph | Quickdraw | Overall | Avg. |
| MoCo V2 (Chen et al., 2020d) | 18.85 | 10.57 | 6.32 | 10.05 | 11.92 | 28.13 | 13.79 | 9.67 | 14.56 | 17.20 |
| SimCLR V2 (Chen et al., 2020c) | 23.51 | 15.42 | 5.29 | 11.80 | 14.74 | 34.03 | 17.17 | 10.88 | 17.32 | 20.69 |
| BYOL (Grill et al., 2020) | 6.21 | 3.48 | 4.27 | 4.45 | 4.65 | 9.60 | 5.09 | 6.02 | 6.49 | 6.90 |
| AdCo (Hu et al., 2021) | 16.16 | 12.26 | 5.65 | 9.57 | 11.36 | 30.77 | 18.65 | 7.75 | 15.44 | 19.06 |
| MAE (He et al., 2021) | 22.38 | 12.62 | 10.50 | 13.51 | 15.17 | 32.60 | 15.28 | 13.43 | 17.85 | 20.44 |
| DARLING (Zhang et al., 2022) | 18.53 | 10.62 | 12.65 | 13.29 | 13.93 | 39.32 | _19.09_ | 10.50 | 18.73 | 22.97 |
| DiMAE (Yang et al., 2022) | _26.52_ | _15.47_ | _15.47_ | _17.72_ | _19.15_ | _42.31_ | 18.87 | _15.00_ | _21.68_ | _25.39_ |
| CycleMAE (ours) | **37.54** | **18.01** | **17.13** | **21.53** | **24.23** | **55.14** | **20.87** | **19.62** | **27.21** | **31.88** |
| Methods | Label Fraction 10% (Full finetuning) | | | | | Label Fraction 100% (Full finetuning) | | | | |
| | Clipart | Infograph | Quickdraw | Overall | Avg. | Clipart | Infograph | Quickdraw | Overall | Avg. |
| MoCo V2 (Chen et al., 2020d) | 32.46 | 18.54 | 8.05 | 15.92 | 19.69 | 64.18 | 27.44 | 25.26 | 33.76 | 38.96 |
| SimCLR V2 (Chen et al., 2020c) | 37.11 | 19.87 | 12.33 | 19.45 | 23.10 | 68.72 | 27.60 | 30.56 | 37.47 | 42.29 |
| BYOL (Grill et al., 2020) | 14.55 | 8.71 | 5.95 | 8.46 | 9.74 | 54.44 | 23.70 | 20.42 | 28.23 | 32.86 |
| AdCo (Hu et al., 2021) | 32.25 | 17.96 | 11.56 | 17.53 | 20.59 | 62.84 | 26.69 | 26.26 | 33.80 | 38.60 |
| MAE (He et al., 2021) | 51.86 | 24.81 | 23.94 | 29.87 | 33.54 | 59.21 | 28.53 | 23.27 | 32.06 | 37.00 |
| DARLING (Zhang et al., 2022) | 35.15 | 20.88 | 15.69 | 21.08 | 23.91 | 72.79 | 32.01 | 33.75 | 41.19 | 46.18 |
| DiMAE (Yang et al., 2022) | _70.78_ | _38.06_ | _27.39_ | _39.20_ | _45.41_ | _83.87_ | **44.99** | _39.30_ | _49.96_ | _56.05_ |
| CycleMAE (Ours) | **74.87** | **38.42** | **28.32** | **40.61** | **47.20** | **85.39** | _44.31_ | **40.03** | **50.46** | **56.58** |

Table 1: The cross-domain generalization results on DomainNet. All of the models are trained on Painting, Real, and Sketch domains of DomainNet and tested on the other three domains. The title of each column indicates the name of the target domain. All the models are pretrained for 1000 epochs before finetuned on the labeled data. Results style: **best**, second best.

| Methods | Label Fraction 1% (Linear evaluation) | | | | | Label Fraction 5% (Linear evaluation) | | | | |
|---|---|---|---|---|---|---|---|---|---|---|
| | Photo | Art. | Cartoon | Sketch | Avg. | Photo | Art. | Cartoon | Sketch | Avg. |
| MoCo V2 (Chen et al., 2020d) | 22.97 | 15.58 | 23.65 | 25.27 | 21.87 | 37.39 | 25.57 | 28.11 | 31.16 | 30.56 |
| SimCLR V2 (Chen et al., 2020c) | 30.94 | 17.43 | _30.16_ | 25.20 | 25.93 | _54.67_ | 35.92 | 35.31 | _36.84_ | 40.68 |
| BYOL (Grill et al., 2020) | 11.20 | 14.53 | 16.21 | 10.01 | 12.99 | 26.55 | 17.79 | 21.87 | 19.65 | 21.47 |
| AdCo (Hu et al., 2021) | 26.13 | 17.11 | 22.96 | 23.37 | 22.39 | 37.65 | 28.21 | 28.52 | 30.35 | 31.18 |
| MAE (He et al., 2021) | 30.72 | 20.78 | 20.78 | 24.52 | 24.89 | 32.69 | 24.61 | 27.35 | 30.44 | 28.77 |
| DARLING (Zhang et al., 2022) | 27.78 | 19.82 | 27.51 | 29.54 | 26.16 | 44.61 | 39.25 | _36.41_ | 36.53 | 39.20 |
| DiMAE (Yang et al., 2022) | _48.86_ | _31.73_ | 25.83 | _32.50_ | _34.23_ | 50.00 | **41.25** | 34.40 | **38.00** | _40.91_ |
| CycleMAE (Ours) | **52.63** | **36.25** | **35.53** | **34.85** | **39.82** | **63.24** | _39.96_ | **42.15** | 36.35 | **45.43** |
| Methods | Label Fraction 10% (Full finetuning) | | | | | Label Fraction 100% (Full finetuning) | | | | |
| | Photo | Art. | Cartoon | Sketch | Avg. | Photo | Art. | Cartoon | Sketch | Avg. |
| MoCo V2 (Chen et al., 2020d) | 44.19 | 25.85 | 33.53 | 24.97 | 32.14 | 59.86 | 28.58 | 48.89 | 34.79 | 43.03 |
| SimCLR V2 (Chen et al., 2020c) | 54.65 | 37.65 | 46.00 | 28.25 | 41.64 | 67.45 | 43.60 | 54.48 | 34.73 | 50.06 |
| BYOL (Grill et al., 2020) | 27.01 | 25.94 | 20.98 | 19.69 | 23.40 | 41.42 | 23.73 | 30.02 | 18.78 | 28.49 |
| AdCo (Hu et al., 2021) | 46.51 | 30.21 | 31.45 | 22.96 | 32.78 | 58.59 | 29.81 | 50.19 | 30.45 | 42.26 |
| MAE (He et al., 2021) | 35.89 | 25.59 | 33.28 | 32.39 | 31.79 | 36.84 | 25.24 | 32.25 | 34.45 | 32.20 |
| DARLING (Zhang et al., 2022) | 53.37 | 39.91 | 46.41 | 30.17 | 42.47 | 68.66 | 41.53 | 56.89 | 37.51 | 51.15 |
| DiMAE (Yang et al., 2022) | _77.87_ | _59.77_ | _57.72_ | **39.25** | _58.65_ | _78.99_ | _63.23_ | _59.44_ | **55.89** | _64.39_ |
| CycleMAE (Ours) | **85.94** | **67.93** | **59.34** | _38.25_ | **62.87** | **90.72** | **75.34** | **69.33** | _50.24_ | **71.41** |

Table 2: The cross-domain generalization results in PACS. Given the experiment for each target domain run respectively, there is no overall accuracy across domains. Thus we report the average accuracy and the accuracy for each domain. The title of each column indicates the name of the domain used as the target. All the models are pretrained for 1000 epochs before finetuned on the labeled data. Results style: **best**, second best.

**+1.79%** and **+0.53%** for DomainNet on average accuracy and gets **+1.41%** and **+0.50%** performance gains on overall accuracy. For PACS, CycleMAE improves the previous methods by **+5.59%** for 1% fraction setting and gets **+4.52%** performance improvements for 5% fraction setting. For 10% and 100% fraction finetuning settings, CycleMAE improves the state-of-the-art method Di-MAE by **+4.22%** and **+7.02%** on average accuracy.

We compare our CycleMAE with previous contrastive-based methods and state-of-the-art generative-based methods, respectively. Previous contrastive learning methods, such as MoCo V2 (Chen et al., 2020d), SimCLR V2 (Chen et al., 2020c), BYOL (Grill et al., 2020), and AdCo (Hu et al., 2021), learn discriminative features using data augmentation to construct positive and negative pairs, which can not explicitly bridge the domain gap by pulling positive pairs together and pushing negative pairs apart. DARLING constructs negative samples for any given queue according to the similarity between different domain samples, but it is still inaccurate and also needs data augmentation to construct positive and negative pairs in the same domain. However, the cross-domain pairs constructed in this way are noisy, which leads to undesirable performance. Compared with contrastive-based methods on UDG setting, our method outperforms all current state-of-the-arts and gets **+9.49%** and **+8.91%** performance improvements on DomainNet on average accuracy using

| $\mathcal{L}_{recons}$ | $\mathcal{L}_{cycle}$ | $\mathcal{L}_{domain}$ | Accuracy |
|:---:|:---:|:---:|:---:|
| ✓ | ✗ | ✗ | 49.42 |
| ✓ | ✓ | ✗ | 54.42 |
| ✓ | ✓ | ✓ | **56.87** |

Table 3: Effectiveness of each proposed component of CycleMAE.

| Loss Design | Accuracy |
|:---|:---:|
| Cosine Distance | 48.87 |
| MMD Loss | 55.21 |
| Domain Contrastive Loss | **56.87** |

Table 4: Comparison of different domain distance regularization loss.

| Cross-domain Pairs | FT | BT | Accuracy |
|:---:|:---:|:---:|:---:|
| | ✗ | ✗ | 49.21 |
| | ✗ | ✓ | 48.78 |
| w/wo Heuristic Pairs | ✓ | ✓ | 51.23 |
| | ✓ | ✗ | **56.87** |

Table 5: Ablation study on heuristic pairs on forward and backward transformation process. The heuristic pair consists of an image and its style-mixed view.

| Decoder Layer | Accuracy |
|:---:|:---:|
| #1 | **56.87** |
| #2 | 56.01 |
| #4 | 55.84 |
| #8 | 54.21 |

Table 6: Comparison of features from different layers for domain contrastive regularization. The decoders consist 8 layers.

linear evaluation protocol and improves the contrastive-based methods by **+23.29%** and **+10.40%** on average accuracy using finetuning evaluation protocol. Furthermore, we find that CycleMAE gets higher performance gains on finetuning evaluation protocol, which is consistent with other unsupervised learning researches (He et al., 2021; Xie et al., 2022).

We also compare our CycleMAE with the state-of-the-art generative-based method DiMAE, which heuristically generates the artificial style-mixed views to construct the cross-domain images. In contrast, CycleMAE utilizes a cycle reconstruction task to construct cross-domain pairs, where we can obtain diverse image pairs from multiple domains. Our cycleMAE shows **+5.08%**, **+6.49%**, **+1.79%**, **+0.53%** gains on 1%, 5%, 10%, 100% DomainNet.

### 3.3 ABLATION STUDY

To further investigate the components of our proposed CycleMAE, we conduct a detailed ablation study on the proposed method. Specifically, we train Vit-Tiny for 100 epochs on the combination of Painting, Real and Sketch training set on DomainNet, and evaluate the model using the linear evaluation protocol on Clipart.

**Effectiveness of Each Component of CycleMAE.** As shown in Tab. 3, we explore the effectiveness of each component in CycleMAE. The cycle reconstruction improves the accuracy by **+5.00%**, and the performance can be further improved with domain contrastive loss by **+2.45%**. Thus, the results verify that the proposed modules can benefit the encoder to learn more domain-invariant features.

**Comparison of Features from Different Layers.** Domain contrastive loss is regularized on the output features of decoders. It raises the question that the features after which layer should be used to obtain the best performances. As shown in Tab. 6, the performance decreases from the first layer to the last layer. As mentioned in (Cao et al., 2022), the semantic information of features increases as the layers go deeper, thus adopting the regularization on the first decoder layer can in turn force the features from the different decoders to share less redundancy. Therefore, we set the regularization after first decoder layer as the default protocol.

**Effectiveness of Domain Distance Regularization Loss.** Domain distance regularization is used to regularize the domain information in different decoders to be different. However, there are many designs of distance regularization. As shown in Tab. 4, the domain contrastive loss regularization gets the best result. That is because the domain contrastive loss regularization pulls the samples from the same domain close and the samples from different domains far away. But other distance metrics only pull the samples from different domains far away without minimizing the intra-domain distance. Thus, the domain contrastive loss improves other domain distance regularization by **+1.66%**.

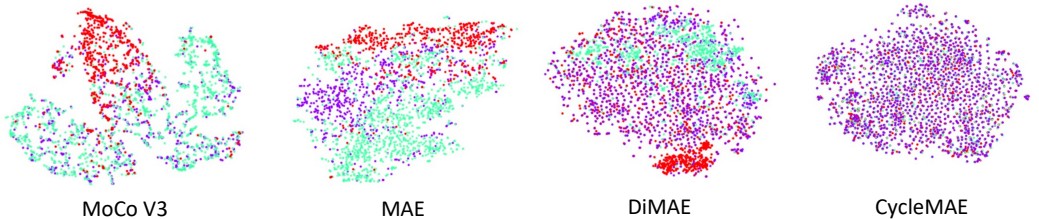

Figure 4: The t-SNE Visualization on the feature distributions with different methods.

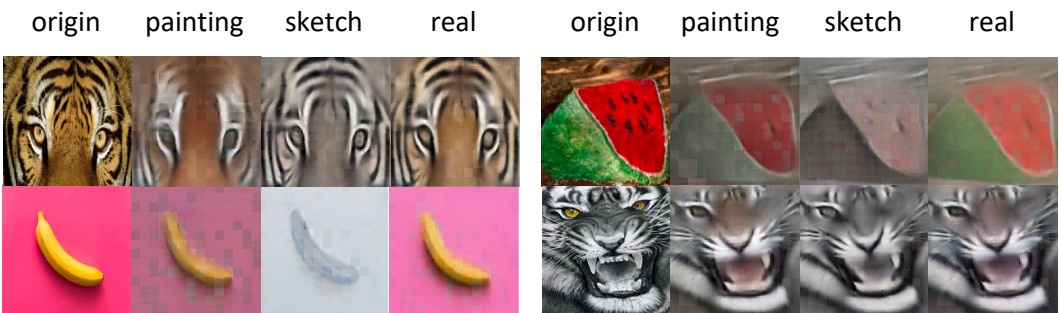

Figure 5: Reconstruction visualization of different decoders.

**Comparison of Cross-domain Pairs for Construction.** Paired data is important but absent in cross-domain datasets. Previous generative-based method (Yang et al., 2022) proposes the heuristic pairs which consist of an original image and its style-mixed view. Compared them with that proposed by (Yang et al., 2022) in Tab. 5, where **FT/BT** denote *forward/backward transformation*, respectively, we demonstrate that CycleMAE gets a **+5.64%** performance gain compared with other situations. Compared with using heuristic pairs in both FT and BT, the optimal setting which uses heuristic pairs only in FT is better because the heuristic images by style mix is not real. However, compared with not using heuristic pairs, using heuristic pairs in FT is important because heuristic pairs in FT can be a good starting point for the encoder in our CycleMAE to learn domain-invariant features and for the decoders to learn domain-specific information.

## 3.4 VISUALIZATION

**Feature Visualization.** We present the feature distribution visualization of MoCo V3, MAE, DiMAE, and CycleMAE in Fig. 4 by t-SNE (Van der Maaten & Hinton, 2008), where the features are part of the combination of Painting, Real, and Sketch training set in DomainNet. From the visualization, we can see the features from different domains in CycleMAE are more uniform than other methods, which indicates that our CycleMAE shows better domain-invariant characteristics.

**Reconstruction Visualization.** In Fig 5, we present the reconstruction results of CycleMAE using ViT-Base for better visualization. From the visualization, we know that our CycleMAE can produce more realistic and diverse pairs that are semantically the same but from different domains.

## 4 CONCLUSIONS

In this paper, we propose the Cycle-consistent Masked AutoEncoder (CycleMAE) to tackle the unsupervised domain generalization problem. CycleMAE designs a cycle reconstruction task to construct cross-domain input-target pairs, thus we can generate more real and diverse image pairs and help to learn a content encoder to extract domain-invariant features. Furthermore, we propose a novel domain contrastive loss to help CycleMAE better disentangle the domain information. Extensive experiments on PACS and DomainNet show that CycleMAE achieves state-of-the-art performance, verifying the effectiveness of our method.

ACKNOWLEDGMENTS

Majority of this work was completed during Haiyang Yang's internship at Sensetime under the mentorship of Feng Zhu. We also extend our gratitude to Qingsong Xie for his contribution to the part of this idea.

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

## 5 APPENDIX

### 5.1 RELATED WORK

**Self-supervised Learning.** Self-supervised learning (SSL) is introduced to learn powerful semantic representations from massive unlabeled data. Recent SSL methods can be divided into two categories: discriminative (Noroozi & Favaro, 2016; Gidaris et al., 2018; Chen et al., 2020b; Grill et al., 2020; He et al., 2020; Chen et al., 2020d; 2021; Zbontar et al., 2021; Caron et al., 2021) and generative methods (Pathak et al., 2016; Larsson et al., 2016; 2017; He et al., 2021). Among the discriminative methods, the early works try to design some auxiliary tasks, like jigsaw puzzle (Noroozi & Favaro, 2016) and rotation prediction (Gidaris et al., 2018), to learn semantic representations. The recent works are mainly based on contrastive loss (Chen et al., 2020b; Grill et al., 2020; He et al., 2020; Chen et al., 2020d; 2021; Zbontar et al., 2021; Caron et al., 2021), which models the similarity and dissimilarity by constructing positive pairs and negative pairs, and learns semantic representations by pulling the positive pairs close and push the negative pairs away.

Generative methods depend on the design of the encoder-decoder structure. Recent methods utilize masked image modeling (MIM) to recover the original images by the masked ones with vision transformer. iGPT (Chen et al., 2020a) predict the pixel value which are from the pixel sequences. MAE (He et al., 2021), a recent state-of-the-art method, recovers the input images based on a few patches of the images for pre-training the autoencoder, which captures semantic representations in this way.

However, these SSL methods only focus on the situation where the training and testing datasets share the same data distribution and there may be performance drops when training and testing datasets exist domain gap. Thus we propose a generative-based method that takes the domain gap into consideration.

**Unsupervised Domain Generalization.** Despite the success, domain generalization still relies on the fully labeled data. To ease the annotation burden, Unsupervised Domain Generalization (UDG) is proposed as a novel generalization task that trains with unlabeled source domains and tests with target domains that serve domain shifts with training domains. Derived from contrastive learning,

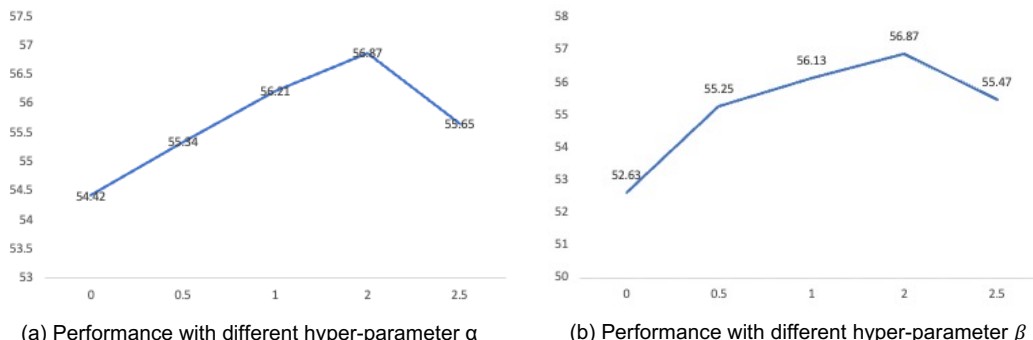

(a) Performance with different hyper-parameter α      (b) Performance with different hyper-parameter $\beta$

Figure 6: Performance with different hyper-parameters. The horizontal axis is the value of hyper-parameters and the vertical axis is the top-1 accuracy.

DARLING (Zhang et al., 2022) incorporates domain information into the contrastive loss by re-weighting domain labels. BrAD (Harary et al., 2021) projects inputs into an auxiliary bridge domain and utilizes contrastive learning in this domain to learn domain-invariant features. Although the above works can eliminate the influence of domain shifts to some extent, their performances are still limited due to the difficulty to well define positive pairs in contrastive learning.

Recently, generative-based methods are proposed to solve the UDG problem. One of the most representative methods is DiMAE (Yang et al., 2022), which establishes an MAE-style (He et al., 2021) generative framework for UDG task. DiMAE contains a content encoder and multiple domain-specific decoders. The input images will be transformed into style-view by the proposed Content-Preserved StyleMix module (Yang et al., 2022), and then be masked randomly. The content encoder extracts domain-invariant features of the style-view counterparts and then the domain specific decoders are designed to recover the reconstruction from the features. In this way, domain-invariant features are learned by the content encoder. Different from DiMAE, our proposed method does not rely on the heuristic pairs which are generated by stylemix and thus can get more realistic cross-domain pairs.

**Cycle-consistency.** Cycle-consistency is a common visual trick to preserve the content unchanged. (Zhou et al., 2016) utilizes the consistency across instances of the same category as a supervised signal to force the model to predict the corresponding relationship between cross instances with the same object. (Yi et al.) and (Zhu et al., 2017) both solve the image-to-image translation problem by cycle-consistency. DualGAN is similar to dual learning and trains both primal and dual GANs at the same time. CycleGAN utilizes cycle-consistency to make the reconstructed images match closely to their original images.

The previous methods we mentioned use the cycle-consistency to remain the object of images unchanged from one image to another image and make two images share the same or similar object. Although we also introduce cycle-consistency to retain the content unchanged, our purpose is not to generate or find another image with the same or similar object. We utilize the cycle-consistency to push the features extracted by the content encoder to retain more content information and thus we can get better domain-invariant features.

## 5.2 MORE EXPERIMENTS

### 5.2.1 HYPER-PARAMETERS SENSITIVITY

As shown in Fig 6 (a), we ablate the loss weight of domain contrastive loss. We observe that the performance is sensitive to the weight of domain contrastive loss, and the Fig 6 (b) shows that cycle reconstruction loss is not sensitive otherwise the weight equals 0. We conjecture the reason is that cycle reconstruction loss, as the key point of the cycle reconstruction task, always works to guarantee the good performance of CycleMAE whatever the weight of it is except 0. But domain contrastive loss acts as a regularization of the cycle reconstruction task, and its weight influences its regularization ability. From the results, we set $\alpha = 2$ and $\beta = 2$.

| method | Label Fraction 1% (linear evaluation) | | | | | Label Fraction 5% (linear evaluation) | | | | |
|---|---|---|---|---|---|---|---|---|---|---|
| | Painting | Real | Sketch | Overall | Avg. | Painting | Real | Sketch | Overall | Avg. |
| ERM | 6.68 | 6.97 | 7.25 | 6.94 | 6.96 | 7.45 | 6.08 | 5.00 | 6.24 | 6.18 |
| MoCo V2 (Chen et al., 2020d) | 11.38 | 14.97 | 15.28 | 14.04 | 13.88 | 20.80 | 24.91 | 21.44 | 23.06 | 22.39 |
| SimCLR V2 (Chen et al., 2020c) | 16.97 | 20.25 | 17.84 | 18.85 | 18.36 | 21.35 | 24.34 | 27.46 | 24.15 | 24.38 |
| BYOL (Grill et al., 2020) | 5.00 | 8.47 | 4.42 | 6.68 | 5.96 | 9.78 | 10.73 | 3.97 | 9.09 | 8.16 |
| AdCo (Hu et al., 2021) | 11.13 | 16.53 | 17.19 | 15.16 | 14.95 | 19.97 | 24.31 | 24.19 | 23.08 | 22.82 |
| MAE(ViT_small) (He et al., 2021) | 17.86 | 24.57 | 19.33 | 21.63 | 20.59 | 24.55 | 30.43 | 26.07 | 27.90 | 27.02 |
| DIUL (Zhang et al., 2022) | 14.45 | 21.68 | 21.30 | 19.59 | 19.14 | 21.09 | 30.51 | 28.49 | 27.48 | 28.19 |
| DiMAE(ViT_small) (Yang et al., 2022) | 20.18 | 30.77 | 20.03 | 25.63 | 23.66 | 27.02 | 39.92 | 26.50 | 33.59 | 31.15 |
| CycleMAE (ViT_tiny) | 21.24 | 25.94 | 15.42 | 22.48 | 20.87 | 23.04 | 29.31 | 19.04 | 25.46 | 23.80 |
| CycleMAE (ViT_small) | **22.85** | 30.38 | **22.31** | 26.63 | 25.18 | **27.64** | **40.24** | **28.71** | **34.38** | **32.20** |
| CycleMAE (ViT_small Supervised Pretrained) | 22.31 | **31.07** | 22.26 | **26.83** | **25.21** | 27.21 | 39.95 | 27.97 | 33.96 | 31.71 |

| method | Label Fraction 10% (full finetuning) | | | | | Label Fraction 100% (full finetuning) | | | | |
|---|---|---|---|---|---|---|---|---|---|---|
| | Painting | Real | Sketch | Overall | Avg. | Painting | Real | Sketch | Overall | Avg. |
| ERM | 9.90 | 9.19 | 5.12 | 8.56 | 8.07 | 31.50 | 40.21 | 24.01 | 34.48 | 31.91 |
| MoCo V2 (Chen et al., 2020d) | 25.35 | 29.91 | 23.71 | 27.37 | 26.32 | 43.42 | 58.61 | 40.38 | 50.66 | 47.47 |
| SimCLR V2 (Chen et al., 2020c) | 24.01 | 30.17 | 31.58 | 28.75 | 28.59 | 46.79 | 62.32 | 51.05 | 55.71 | 53.39 |
| BYOL (Grill et al., 2020) | 9.50 | 10.38 | 4.45 | 8.92 | 8.11 | 34.02 | 46.48 | 24.82 | 38.59 | 35.11 |
| AdCo (Hu et al., 2021) | 23.35 | 29.98 | 27.57 | 27.65 | 26.97 | 43.55 | 61.42 | 51.23 | 54.37 | 52.07 |
| MAE (ViT_small) (He et al., 2021) | 41.24 | 54.68 | 39.41 | 47.82 | 45.11 | 53.13 | 68.51 | 48.86 | 60.21 | 56.83 |
| DARLING (Zhang et al., 2022) | 25.90 | 33.29 | 30.77 | 30.72 | 29.99 | 49.64 | 63.77 | 54.31 | 57.91 | 55.91 |
| DiMAE (ViT_small) (Yang et al., 2022) | 50.73 | 64.89 | 55.41 | 59.01 | 57.01 | 70.48 | 82.79 | 72.10 | 77.18 | 75.12 |
| CycleMAE (ViT_tiny) | 43.42 | 57.75 | 41.84 | 50.51 | 47.67 | 56.92 | 72.52 | 53.28 | 64.24 | 60.91 |
| CycleMAE (ViT_small) | 52.81 | 67.13 | **56.37** | 60.95 | 58.77 | **72.01** | **84.95** | 73.84 | 79.08 | 76.93 |
| CycleMAE (ViT_small Supervised Pretrained) | **53.40** | **67.24** | 55.72 | **61.04** | **58.79** | 71.93 | 84.87 | **74.62** | **79.18** | **77.14** |

Table 7: Results of the cross-domain generalization on DomainNet. All of the models are trained on Clipart, Infograph, Quickdraw domains of DomainNet and tested on the other three domains. The title of each column indicates the name of the domain used as target. All the models are pretrained for 1000 epoches before finetuned on the labeled data. Results style: **best**, second best.

### 5.2.2 EXPERIMENTS ON OPPOSITE SETTING OF DOMAINNET

We showed part of our results in the main text where we train our model on Painting, Real, and Sketch, and evaluate the generalization ability of our model on Clipart, Infograph, and Quickdraw. In this section, we train our model on Clipart, Infograph, and Quickdraw, and evaluate it on Painting, Real, and Sketch. Same as we mentioned in the main text, we exactly follow the all correlated setting proposed by DARLING (Zhang et al., 2022).

The results are presented in Tab 7. Our CycleMAE still achieves a good performance. We achieve a improvement by **+1.00%** and **+0.79%** on overall accuracy and **+1.52%** and **+1.05%** on average accuracy for 1% and 5% fraction setting. Specifically, for 10% fraction and 100% fraction setting, our CycleMAE improves the state-of-the-art methods by **+1.76%** and **+1.81%** on average accuracy and **+1.94%** and **+1.90%** on overall accuracy.

### 5.3 EXPERIMENTS WITH VIT_TINY BACKBONE

In this section, we use a smaller backbone, ViT_tiny, to illustrate the effectiveness of our proposed CycleMAE. We still follow the all correlated setting of DARLING (Zhang et al., 2022).

### 5.3.1 EXPERIMENTS ON DOMAINNET

In this section, we use the ViT_tiny as the backbone of our proposed CycleMAE and evaluate the performance on DomainNet with two tasks. The first task is training our model on Painting, Real and Sketch, and evaluating the generalization ability of our model on Clipart, Infograph and Quickdraw (Painting, real, sketch → Clipart, Infograph, Quickdraw). The second task is training our model on Clipart, Infograph and Quickdraw, and evaluating it on Painting, Real and Sketch (Clipart, Infograph, Quickdraw → Painting, real, sketch).

| | Label Fraction 1% | | | | | Label Fraction 5% | | | | |
|---|---|---|---|---|---|---|---|---|---|---|
| method | Clipart | Infograph | Quickdraw | Overall | Avg. | Clipart | Infograph | Quickdraw | Overall | Avg. |
| ERM | 6.54 | 2.96 | 5.00 | 4.75 | 4.83 | 10.21 | 7.08 | 5.34 | 6.81 | 7.54 |
| MoCo V2 (Chen et al., 2020d) | 18.85 | 10.57 | 6.32 | 10.05 | 11.92 | 28.13 | 13.79 | 9.67 | 14.56 | 17.20 |
| SimCLR V2 (Chen et al., 2020c) | 23.51 | 15.42 | 5.29 | 11.80 | 14.74 | 34.03 | 17.17 | 10.88 | 17.32 | 20.69 |
| BYOL (Grill et al., 2020) | 6.21 | 3.48 | 4.27 | 4.45 | 4.65 | 9.60 | 5.09 | 6.02 | 6.49 | 6.90 |
| AdCo (Hu et al., 2021) | 16.16 | 12.26 | 5.65 | 9.57 | 11.36 | 30.77 | 18.65 | 7.75 | 15.44 | 19.06 |
| MAE (ViT_small) (He et al., 2021) | 22.38 | 12.62 | 10.50 | 13.51 | 15.17 | 32.60 | 15.28 | 13.43 | 17.85 | 20.44 |
| DARLING (Zhang et al., 2022) | 18.53 | 10.62 | 12.65 | 13.29 | 13.93 | 39.32 | 19.09 | 10.50 | 18.73 | 22.97 |
| DiMAE (ViT_small) (Yang et al., 2022) | 26.52 | 15.47d | _15.47_ | 17.72 | 19.15 | 42.31 | 18.87 | 15.00 | 21.68 | 25.39 |
| CycleMAE (ViT_tiny) | 28.04 | 14.23 | 14.34 | 17.10 | 18.87 | 37.82 | 18.44 | 16.12 | 21.19 | 24.13 |
| CycleMAE (ViT_small) | _37.54_ | _18.01_ | **17.13** | **21.53** | **24.23** | _55.14_ | _20.87_ | **19.62** | **27.21** | _31.88_ |
| CycleMAE (ViT_small Supervised Pretrained) | **39.02** | **18.63** | 14.60 | _20.70_ | 24.08 | **56.27** | **22.22** | _17.97_ | _26.96_ | **32.15** |
| | Label Fraction 10% | | | | | Label Fraction 100% | | | | |
| method | Clipart | Infograph | Quickdraw | Overall | Avg. | Clipart | Infograph | Quickdraw | Overall | Avg. |
| ERM | 15.10 | 9.39 | 7.11 | 9.36 | 10.53 | 52.79 | 23.72 | 19.05 | 27.19 | 31.85 |
| MoCo V2 (Chen et al., 2020d) | 32.46 | 18.54 | 8.05 | 15.92 | 19.69 | 64.18 | 27.44 | 25.26 | 33.76 | 38.96 |
| SimCLR V2 (Chen et al., 2020c) | 37.11 | 19.87 | 12.33 | 19.45 | 23.10 | 68.72 | 27.60 | 30.56 | 37.47 | 42.29 |
| BYOL (Grill et al., 2020) | 14.55 | 8.71 | 5.95 | 8.46 | 9.74 | 54.44 | 23.70 | 20.42 | 28.23 | 32.86 |
| AdCo (Hu et al., 2021) | 32.25 | 17.96 | 11.56 | 17.53 | 20.59 | 62.84 | 26.69 | 26.26 | 33.80 | 38.60 |
| MAE (ViT_small) (He et al., 2021) | 51.86 | 24.81 | 23.94 | 29.87 | 33.54 | 59.21 | 28.53 | 23.27 | 32.06 | 37.00 |
| DARLING (Zhang et al., 2022) | 35.15 | 20.88 | 15.69 | 21.08 | 23.91 | 72.79 | 32.01 | 33.75 | 41.19 | 46.18 |
| DiMAE (ViT_small) (Yang et al., 2022) | 70.78 | 38.06 | 27.39 | 39.20 | 45.41 | 83.87 | **44.99** | 39.30 | 49.96 | 56.05 |
| CycleMAE (ViT_tiny) | 61.98 | 33.14 | 21.87 | 33.18 | 39.00 | 72.25 | 38.47 | 24.25 | 37.99 | 44.99 |
| CycleMAE (ViT_small) | _74.87_ | _38.42_ | _28.32_ | _40.61_ | _47.20_ | _85.39_ | 44.31 | _40.03_ | _50.46_ | _56.58_ |
| CycleMAE (ViT_small Supervised Pretrained) | **75.93** | **39.11** | **29.19** | **41.47** | **48.08** | **86.40** | _44.93_ | **40.56** | **51.12** | **57.30** |

Table 8: Results of the cross-domain generalization on DomainNet. All of the models are trained on Painting, Real, Sketch domains of DomainNet and tested on the other three domains. The title of each column indicates the name of the domain used as target. All the models are pretrained for 1000 epochs before finetuned on the labeled data. Results style: **best**, second best.

**Painting, real, sketch → Clipart, Infograph, Quickdraw.** We present the results of this task in Tab. 8. From the results, our proposed CycleMAE achieves a competitive performance even with the ViT_tiny. For 1% and 5% fraction setting, our CycleMAE get a similar result compared with the previous state-of-the-art method DiMAE (Yang et al., 2022), although ViT_tiny is smaller than the backbone used in DiMAE. Except DiMAE with ViT_small, our CycleMAE improves other previous methods by **+3.70%**, **+1.16%**, **+5.09%** on average accuracy for 1%, 5%, 10% fraction setting.

**Clipart, Infograph, Quickdraw → Painting, real, sketch.** The results of this task are shown in Tab. 7. For a fair comparison, we do not compare our CycleMAE with ViT_tiny with DiMAE which use the ViT_small as the backbone. Compared with previous methods except DiMAE, our CycleMAE improve the average accuracy by **+0.28%**, **+2.56%** and **+4.08%** for 1%, 10% and 100% fraction setting.

### 5.3.2 EXPERIMENT ON PACS

We also use the PACS to evaluate the effectiveness of our CycleMAE. We present our results in S-Table 9. In this setting, our CycleMAE achieves a better performance than previous works on most tasks and gets significant gains over DiMAE, DARLING and other SSL methods on average accuracy. Compared with state-of-the-art methods, our CycleMAE improves the accuracy by **+2.46%**, **+0.60%** on average for 1%, 5% fraction setting, respectively. And for 10% and 100% fraction setting, compared with other state-of-the-art methods except DiMAE, we get **+14.56%** and **+10.80%** performance gains on average accuracy.

### 5.4 VISUALIZATION

We visualize more reconstruction results of CycleMAE using ViT-base in Fig 7. We can know from the visualization that the cross-domain pairs generated by CycleMAE have good content-consistency.

| method | Label Fraction 1% | | | | | Label Fraction 5% | | | | |
|---|---|---|---|---|---|---|---|---|---|---|
| | Photo | Art. | Cartoon | Sketch | Avg. | Photo | Art. | Cartoon | Sketch | Avg. |
| MoCo V2 (Chen et al., 2020d) | 22.97 | 15.58 | 23.65 | 25.27 | 21.87 | 37.39 | 25.57 | 28.11 | 31.16 | 30.56 |
| SimCLR V2 (Chen et al., 2020c) | 30.94 | 17.43 | 30.16 | 25.20 | 25.93 | 54.67 | 35.92 | 35.31 | _36.84_ | 40.68 |
| BYOL (Grill et al., 2020) | 11.20 | 14.53 | 16.21 | 10.01 | 12.99 | 26.55 | 17.79 | 21.87 | 19.65 | 21.47 |
| AdCo (Hu et al., 2021) | 26.13 | 17.11 | 22.96 | 23.37 | 22.39 | 37.65 | 28.21 | 28.52 | 30.35 | 31.18 |
| MAE(ViT_small) (He et al., 2021) | 30.72 | 23.54 | 20.78 | 24.52 | 24.89 | 32.69 | 24.61 | 27.35 | 30.44 | 28.77 |
| DARLING (Zhang et al., 2022) | 27.78 | 19.82 | 27.51 | 29.54 | 26.16 | 44.61 | 39.25 | 36.41 | 36.53 | 39.20 |
| DiMAE (ViT_small) (Yang et al., 2022) | 48.86 | 31.73 | 25.83 | 32.50 | 34.23 | 50.00 | **41.25** | 34.40 | **38.00** | 40.91 |
| CycleMAE (ViT_tiny) | 51.89 | 33.86 | 30.74 | 30.27 | 36.69 | 58.28 | 34.82 | 38.72 | 34.20 | 41.51 |
| CycleMAE (ViT_small) | _52.63_ | _36.25_ | _35.53_ | **34.85** | _39.82_ | _63.24_ | _39.96_ | _42.15_ | 36.35 | _45.43_ |
| CycleMAE (ViT_small Supervised Pretrained) | **53.71** | **38.13** | **36.05** | _34.48_ | **40.59** | **64.17** | 39.45 | **43.98** | 35.50 | **45.78** |
| method | Label Fraction 10% | | | | | Label Fraction 100% | | | | |
| | Photo | Art. | Cartoon | Sketch | Avg. | Photo | Art. | Cartoon | Sketch | Avg. |
| MoCo V2 (Chen et al., 2020d) | 44.19 | 25.85 | 33.53 | 24.97 | 32.14 | 59.86 | 28.58 | 48.89 | 34.79 | 43.03 |
| SimCLR V2 (Chen et al., 2020c) | 54.65 | 37.65 | 46.00 | 28.25 | 41.64 | 67.45 | 43.60 | 54.48 | 34.73 | 50.06 |
| BYOL (Grill et al., 2020) | 27.01 | 25.94 | 20.98 | 19.69 | 23.40 | 41.42 | 23.73 | 30.02 | 18.78 | 28.49 |
| AdCo (Hu et al., 2021) | 46.51 | 30.21 | 31.45 | 22.96 | 32.78 | 58.59 | 29.81 | 50.19 | 30.45 | 42.26 |
| MAE(ViT_small) (He et al., 2021) | 35.89 | 25.59 | 33.28 | 32.39 | 31.79 | 36.84 | 25.24 | 32.25 | 34.45 | 32.20 |
| DARLING (Zhang et al., 2022) | 53.37 | 39.91 | 46.41 | 30.17 | 42.47 | 68.66 | 41.53 | 56.89 | 37.51 | 51.15 |
| DiMAE (ViT_small) (Yang et al., 2022) | 77.87 | 59.77 | 57.72 | **39.25** | 58.65 | 78.99 | 63.23 | 59.44 | **55.89** | 64.39 |
| CycleMAE (ViT_tiny) | 83.27 | 56.42 | 53.58 | 34.85 | 57.03 | 89.46 | 60.47 | 60.84 | 37.04 | 61.95 |
| CycleMAE (ViT_small) | _85.94_ | _67.93_ | _59.34_ | _38.25_ | _62.87_ | _90.72_ | _75.34_ | _69.33_ | 50.24 | _71.41_ |
| CycleMAE (ViT_small Supervised Pretrained) | **87.02** | **68.31** | **60.20** | 37.40 | **63.23** | **92.17** | **77.69** | **70.56** | _51.07_ | **72.87** |

Table 9: Results of the cross-domain generalization setting on PACS. Given the experiment for each target domain runs respectively, there is no overall accuracy across domains. Thus we report the average accuracy and the accuracy for each domain. The title of each column indicates the name of the domain used as target. All the models are pretrained for 1000 epochs before finetuned on the labeled data. Results style: **best**, _second best_.

## 5.5 BROADER IMPACT

We propose an effective generative-based unsupervised domain generalization method and we can get a more realistic cross-domain pairs from model outputs. However, in our experiments, there are a potential issue that we should consider to remedy in the future. The issue is that our experiments rely on many GPUs to pretrain and test, which may consume a lot of electricity. And we know that using too much electricity can cause pollution which may influence our world.

Origin          Painting          Real          Sketch

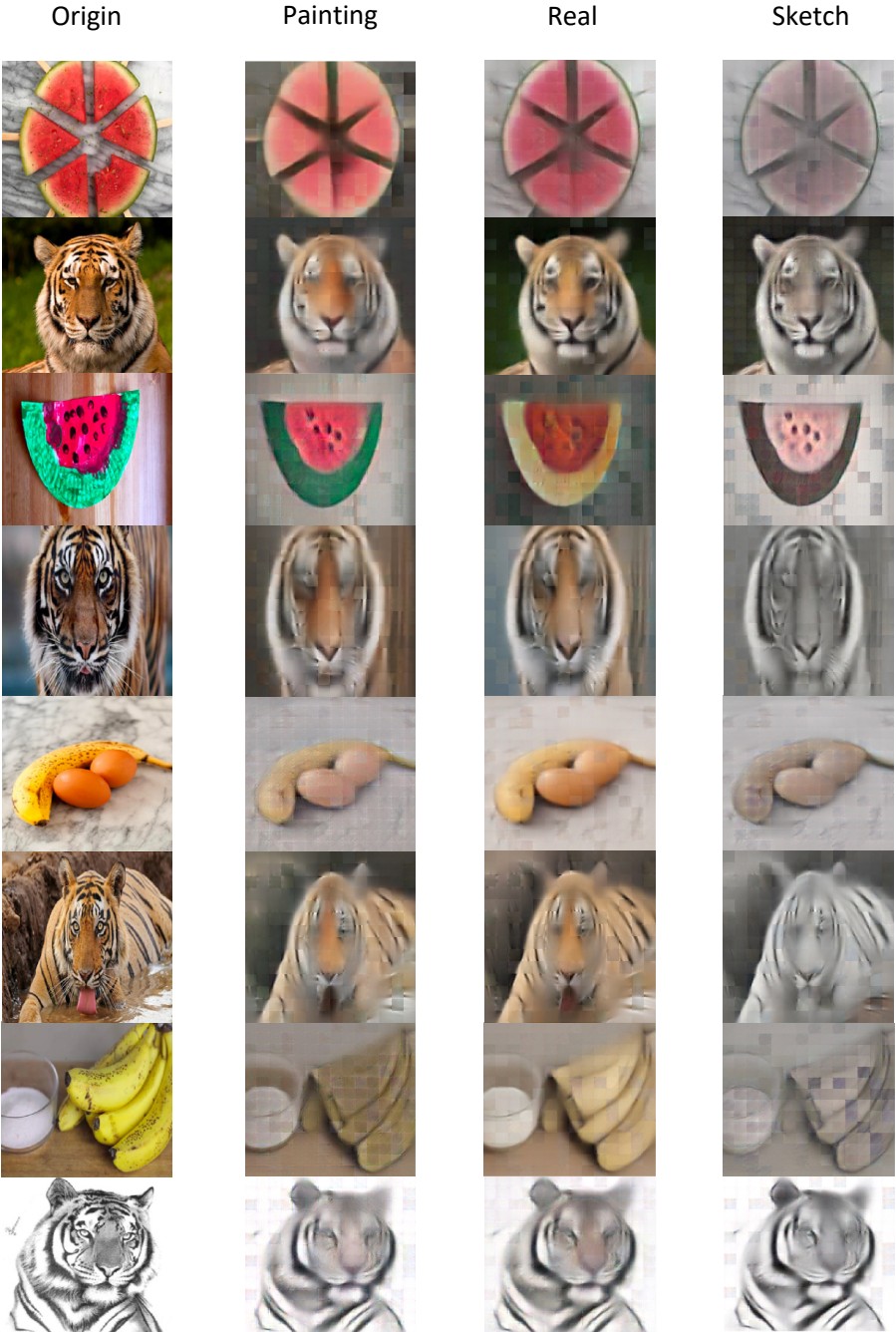

Figure 7: Reconstruction visualization of different decoders.

