# OpenReview forum: " Cycle-consistent Masked AutoEncoder for Unsupervised Domain Generalization"
_ICLR.cc/2023/Conference — ICLR 2023 poster_

### Official Review · Reviewer_7ZLc · 2022-10-20

**Confidence:** 3
**Correctness:** 3
**Technical Novelty And Significance:** 3
**Empirical Novelty And Significance:** 3
**Recommendation:** 6

**Clarity, Quality, Novelty And Reproducibility:**

+ The paper is well-written and easy to follow.
+ If compared with previous work DiMAE, the novelty of this paper is somehow limited. Based on the previous reconstruction loss, this paper proposes to further include a cycle-consistent one. The similar story has repeated frequently in the field.
+ Since the unsupervised domain generalization setting is relatively new, the reviewer finds that most of the published methods do not have open implementations. To this end, the reviewer is not sure whether the experiments are conducted in a fair environment and whether there are some tricks hinder the reproduction.


**Strength And Weaknesses:**

Issues regarding the fair experimental comparisons.
+ To the reviewer’s knowledge, some previous works such as DiMAE use a supervised pre-trained model while this paper adapts an unsupervised pre-trained model on ImageNet. To this end, could the authors explain how much will the different pre-trained models affect the final performance? Meanwhile, pre-trained models using different pretraining recipes (e.g., number of training epochs) are also different starting points for downstream tasks. Overall, the reviewer hopes the authors to make sure the experimental comparisons are fair in terms of the pre-trained model.
***
Issues regarding the methods.
+ Could the authors clarify the extra training overhead they have introduced in their framework? Because more decoders and forward passes are appended upon previous approach.
+ As claimed in the paper that, the domain contrastive loss encourages more diverse domain-specific styles to be learned in the decoder. The reviewer is interested that why such constraints should be placed in the early layer of the decoder since the higher layer closer to the pixel space might be more responsible to encoder style information?
+ The unsupervised domain generalization is a new setting in the field. The reviewer believes that reporting the performance in the unsupervised source domain is also of significance. Could the authors provide such information?


**Summary Of The Paper:**

This paper focuses on the setting of unsupervised domain generalization, where the multiple source domains do not have label supervision and the target domain is unseen. Based on the previous work (i.e., DiMAE) that applies the masked-reconstruction-modeling as a self-supervised task, this paper further proposes two more objective functions: the cycle reconstruction loss and domain contrastive loss. The first one loss transforms the generated image back to original image domain and the second one loss performs a contrastive learning across the features from different decoders. The authors have conducted experiments on the DomainNet and PACS and achieves some improvements over the previous methods.

**Summary Of The Review:**

This paper proposes to use a cycle-reconstruction loss and domain contrastive loss in the unsupervised domain generalization. Consistent performance gains are detected on two widely-used benchmarks. Currently, the reviewer’s concerns are two-fold. On the one hand, the authors should make sure that the experimental comparisons are fair and reproducible. On the other hand, the two new losses are kind of limited in the terms of novelty, which might be lower than the bar to be accepted to a venue like ICLR.

---

### Official Review · Reviewer_BvPn · 2022-10-24

**Confidence:** 3
**Clarity, Quality, Novelty And Reproducibility:** The paper is well-written and novel.
**Correctness:** 3
**Technical Novelty And Significance:** 3
**Empirical Novelty And Significance:** 3
**Recommendation:** 6

**Strength And Weaknesses:**

Pro:

1.  Previous method DiVAE only considers the reconstruction loss ||y_m-x_m||. The proposed loss considers how to utilize the information of y_n and use cycle consistency to further help the task, i.e., adding a backward transformation process for other domain labels $n\neq m$.

2. The paper also notices that the generated images in other domains $$y_n$$ may be trivial such that all images are of the same style. To encourage diverse domain information, they propose a domain contrastive loss.

3. The results are promising and the paper is well-written.

Cons:

1.  Since the model adds a backward process compared to DiVAE, I'm wondering how much training time is increased. It would be good to have a running time comparison.

Comments:

The proposed domain contrastive loss encourages the outputs to have different domain styles but there is no supervised loss to encourage it to be exactly the domain style i. Is it possible that the outputs of different domain-specific decoders are out of the given domains and harm the performance?



**Summary Of The Paper:**

This paper proposes to use cycle consistency to help learn domain-invariant features for unsupervised domain generalization.

**Summary Of The Review:**

The paper proposes to use cycle consistency to help learn the domain-invariant features and the results are promising.

---

### Official Review · Reviewer_QvTm · 2022-10-27

**Confidence:** 3
**Clarity, Quality, Novelty And Reproducibility:** The quality, clarity and originality …
**Correctness:** 2
**Technical Novelty And Significance:** 2
**Empirical Novelty And Significance:** 2
**Recommendation:** 6

**Strength And Weaknesses:**

# Strength

1. The motivation of introducing CycleGAN, MaskAutoEncoder, and contrastive learning sounds reasonable.

2. The experimental results look solid.

3. The writing is good, and the structure is easy to follow.

# Weaknesses

1. The novelty of this paper seems marginal. The main modules of the proposed method, including CycleGAN, MaskAutoEncoder, and contrastive learning, are all existing famous works. It seems that the authors simply combine these works into UDA.

2. The visualization results in Figure 5 are unsatisfied, especially the sketch ones. It is highly suggested to release more visualization results to demonstrate the effectiveness of the reconstrcution process.

3. I notice that the authors use Vit-small as the backbone network. Do all the other comparison methods use this network? If not, it is suggested to compare the backbones of different methods, including training time, model size, and inference speed.

4. The proposed method needs to be pre-trained for 1000 epochs, which may easy to cause overfitting. It is suggested to add a set of experiments to fully prove the generalization of the proposed method.


**Summary Of The Paper:**

This paper aims to address unsupervised domain generalization via a proposed cycle-consistent masked autoencoder, which includes a CycleGAN, MaskAutoEncoder, and contrastive learning. The authors claim that the poposed method can significantly improve state-of-the-art performances, which are demonstrated by the experimental results to some extent.

**Summary Of The Review:**

Please see [Strength And Weaknesses].

---

### Decision · Program_Chairs · 2023-01-20

**Decision:**

Accept: poster

**Justification For Why Not Higher Score:**

The current paper is believed by the reviewers and the AC to be incremental in its technical solutions. While the empirical performance is good, which is acceptable, there is no evidence that it shall be presented as spotlights.

**Justification For Why Not Lower Score:**

All three reviewers acknowledged that their concerns were addressed well. The paper has its merits in tackling a challenging UDG problem with a reasonable solution that does not rely on paired data. This is a solid empirical work that is a meaningful addition to the conference. Nonetheless, since this is a highly selective conference, it is not a great loss if this paper is rejected.

**Metareview: Summary, Strengths And Weaknesses:**

This paper studies an interesting problem, unsupervised domain generalization (UDG), in which no paired data is required to bridge different domains. This problem is relatively under-explored, but it is meaningful in the DG context. The proposed approach is a complex combination of well-established methods, such as MAE, CycleGAN, and StyleMix. While the method is more engineering-oriented, the experimental performance is very good, which can be seen as the biggest merit of this paper. Reviews were with mixed ratings at the beginning. After rebuttal, all three reviewers acknowledged that their concerns were addressed reasonably well. AC read the paper and oversaw the reviewing process, and agreed that this paper has its merits that outweigh its flaws, and the current version is slightly above the acceptance threshold. Authors should incorporate the revised material into the future version.

**Note From Pc:**

if the above contains the word "oral" or "spotlight" please see: "oral" presentation means -> notable-top-5% and "spotlight" means -> notable-top-25%. As stated in our emails, we are disassociating presentation type from AC recommendations